# Virtual Dental Articulation Using Computed Tomography Data and Motion Tracking

**DOI:** 10.3390/bioengineering10111248

**Published:** 2023-10-25

**Authors:** Ting-Han Chou, Shu-Wei Liao, Jun-Xuan Huang, Hsun-Yu Huang, Hien Vu-Dinh, Hong-Tzong Yau

**Affiliations:** 1Department of Stomatology, Ditmanson Medical Foundation Chia-Yi Christian Hospital, Chiayi 600, Taiwan; 07314@cych.org.tw (T.-H.C.); cych07114@gmail.com (H.-Y.H.); 2Department of Mechanical Engineering, Advanced Institute of Manufacturing with High-Innovation, National Chung Cheng University, Chiayi 621, Taiwan; shuweiliao645@gmail.com (S.-W.L.); gs1458@gmail.com (J.-X.H.); vuhien260697@alum.ccu.edu.tw (H.V.-D.); 3School of Dentistry Kaohsiung, Medical University Kaohsiung, Kaohsiung 807, Taiwan

**Keywords:** TMJ, virtual articulation, digital dentistry, optical tracking, occlusion analysis

## Abstract

Dental articulation holds crucial and fundamental importance in the design of dental restorations and analysis of prosthetic or orthodontic occlusions. However, common traditional and digital articulators are difficult and cumbersome in use to effectively translate the dental cast model to the articulator workspace when using traditional facebows. In this study, we have developed a personalized virtual dental articulator that directly utilizes computed tomography (CT) data to mathematically model the complex jaw movement, providing a more efficient and accurate way of analyzing and designing dental restorations. By utilizing CT data, Frankfurt’s horizontal plane was established for the mathematical modeling of virtual articulation, eliminating tedious facebow transfers. After capturing the patients’ CT images and tracking their jaw movements prior to dental treatment, the jaw-tracking information was incorporated into the articulation mathematical model. The validation and analysis of the personalized articulation approach were conducted by comparing the jaw movement between simulation data (virtual articulator) and real measurement data. As a result, the proposed virtual articulator achieves two important functions. Firstly, it replaces the traditional facebow transfer process by transferring the digital dental model to the virtual articulator through the anatomical relationship derived from the cranial CT data. Secondly, the jaw movement trajectory provided by optical tracking was incorporated into the mathematical articulation model to create a personalized virtual articulation with a small Fréchet distance of 1.7 mm. This virtual articulator provides a valuable tool that enables dentists to obtain diagnostic information about the temporomandibular joint (TMJ) and configure personalized settings of occlusal analysis for patients.

## 1. Introduction

The dental industry has undergone significant changes in recent years with the rapid growth of digital dentistry [1,2,3]. Advancements such as cone beam computed tomography (CBCT) [4], intra-oral scanners [5], dental CAD/CAM software [6,7], 5-axis milling machines [8], and 3D printers [9,10] have enabled almost all dental prostheses to be manufactured by computerized systems. However, careful analysis of the occlusion and matching between opposing teeth is required when creating single crowns or multi-unit bridges. Traditionally, laboratory technicians have used mechanical articulators to accomplish this task. 

First known by the work of Dr. Daniel T. Evens in 1840 [11], the articulator was subsequently developed and became widespread in use. A condylar path is defined with an inclination [12], which causes the condylar ball to move with a fixed inclination. In 1899, an instrument known as a facebow was first introduced to determine specific information about the human skull, particularly the jaw relationship, and then transferred to the articulators [13]. This helps ensure that the upper jaw and the lower jaw are positioned correctly. Following these developments, improved versions of the articulator emerged, such as Gysi’s adaptable articulator in 1910 [14] and Szentpetery’s digital articulator in 1997 [15]. In 2002, Gärtner and Kordass [16] recorded the jaw movement by a jaw motion analyzer (JMA) and used it to simulate the static and dynamic occlusion contacts in the computer system DentCAM.

For decades, a number of different methods have been developed for jaw movement assessment. An ultrasound device was developed quite early for jaw movement measurement [17]. Electromagnetic sensors were also developed and used to analyze the 3D mandibular movement [18]. Pinheiro et al. [19] developed an early simple optical method to track the relative jaw movement, followed by more sophisticated optical tracking methods [20,21,22] in recent years. With the development of various methods and devices for the tracking of mandible movement, most of the research in the past focused on the recording of the mandible trajectories during the protrusion or lateral excursions. However, there has been limited attention given to the (mathematical) modeling of the articulation and its integration with the CT data and the tracking data for the customization of a patient-specific dental articulation model. Just like programming a robot requires the mathematical modeling of the kinematics, to fully simulate the mastication or occlusion functions, it requires the building of a patient-specific articulation model.

A complete digital workflow now involves the use of digital or virtual articulators for restoration design and even orthodontic occlusion analysis [9,23]. In our previous work [24], we proposed a mathematical modeling of dental articulation and utilized 3D optical tracking to verify its accuracy. The kinematics of a universal mechanical articulator were analyzed to simulate the relative jaw movement between the maxilla and mandible. The developed digital version successfully imitates jaw motion, simulating occlusion paths, and detecting tooth collisions for applications in dental occlusion analysis. Despite the integration of digital articulators in the occlusion analysis for prosthesis design, there is still a tedious process involved in transferring the digital dental cast model to the workspace of a digital articulator. Traditionally, this has been achieved through the use of a mechanical facebow [25,26]. Similarly, in the virtual articulation process, it is crucial to ensure that this relationship is correctly transferred to the virtual articulator, to ensure the simulation of articulating motion is consistent with that of a real human body. Additionally, each person has unique occlusion and articulation motions [27]. Incorporating this personalized motion into virtual articulation models remains a challenge. In this paper, we address these issues by developing a personalized virtual articulation model, utilizing CT data and personalized jaw movement tracking.

The aim of this study is to develop a virtual articulator capable of reproducing personalized jaw movements and establish a digital workflow for creating a virtual articulation by utilizing jaw movement and information obtained from CT scans. The patient’s jaw movement is tracked before dental treatment, and this information is integrated into the articulation mechanism to construct a personalized mathematical model of jaw movement. The development of the virtual articulator involves the utilization of C++ and OpenGL libraries, with the application of VTK and ITK libraries for jaw extraction software. This model serves as a valuable tool for dentists and dental technicians in prosthesis design or treatment planning, particularly in cases where the patient is not physically present at the clinic. Furthermore, the mathematical model enables precise control over mandibular movement without the need for a mechanical or digital articulator. This eliminates the cumbersome process of employing a facebow for transferring digital cast models to the articulator space. The workflow for constructing the personalized virtual articulation is depicted in Figure 1.

## 2. Materials and Methods

### 2.1. Mathematical Modeling of Virtual Articulation

Mandibular movement is determined by the activities of TMJs, with the condyles being crucial components in expressing these movements [28]. The precise position of the condyles can be determined from CT data. Therefore, the approach to model personalized virtual articulation through mathematical analysis using CT data is carried out based on the rotation and translation of the lower jaw relative to the upper jaw [24]. This involves matching the actual condyles with the corresponding virtual condyles in the established modeling. The rotational and translational movements of the mandible occur simultaneously in reality and are complex. However, they can be modeled through the utilization of the Bennett angle joint and condyle inclination joint within an articulator. Assuming the midpoint of the line connecting the two condyles is designated as joint 0, the Bennett angle joint and the condyle inclination joint (on the right side) are labeled as joints 1 and 2, respectively. The end-effector (incisal tip, point P) can be determined through transformations from the midpoint coordinate (origin of the world coordinate) to the Bennett angle joint coordinate ($T_{0}^{1}$), and then to the condyle inclination joint coordinate ($T_{1}^{2}$). These transformations can be represented as Equations (1) and (2):(1)T01=100 010P0 1001 000110000cos⁡θsag−sin⁡θsag00sin⁡θsagcos⁡θsag00001cos⁡θ1−sin⁡θ100sin⁡θ1cos⁡θ10000100001
(2)T12=100 010P1 2001 000110000cos⁡θ2−sin⁡θ200sin⁡θ2cos⁡θ200001
where P0 1, P1 2 represent translations from the world coordinate to the two joints. $\theta_{sag}$, $\theta_{1}$, and $\theta_{2}$ are angles between the horizontal plane and the condyle inclination plane, Bennett angle joint, and the condyle inclination adjuster, respectively. The same approach can be employed for the left side. Additionally, the condylar path has been constructed by collecting points on the $P_{Ri}^{\;}$ path ($i\;{\in N}^{}$), as defined by Equation (3):(3)$${\left[P_{Ri}^{\;}\right]}^{0}={{\left(T_{1}^{2}T_{0}^{1}\right)}^{-1}[P_{Ri}^{\;}]}^{2}=T_{1}^{0}T_{2}^{1}{\left[P_{Ri}^{\;}\right]}^{2}$$

For the inverse kinematic analysis, we consider the coordinates of the condyle positions in the initial (centric relation) state as ${\left[x_{R},y_{R},z_{R}\right]}^{T}$ and ${\left[x_{L},y_{L},z_{L}\right]}^{T}$ for the right and left sides, respectively. After the jaw movement, the corresponding coordinates become ${\left[x_{R}^{\prime },y_{R}^{\prime },z_{R}^{\prime }\right]}^{T}$ and ${\left[x_{L}^{\prime },y_{L}^{\prime },z_{L}^{\prime }\right]}^{T}$. Considering the right lateral excursion, the positions of the incisal tip in the initial and final poses are defined as point P and point Q, respectively. These positions are linked by the transformation matrix ($T_{0}^{R}$), which maps coordinates from the world coordinate system $\left\{X^{0},\;Y^{0},\;Z^{0}\right\}$ to the right reference coordinate system $\left\{X^{R},\;Y^{R},\;Z^{R}\right\}$. As a result, the following relationships hold:(4)$${\left[P\right]}^{R}=T_{0}^{R}{\left[P\right]}^{0}$$
(5)$${\left[Q\right]}^{R}=T_{0}^{R}{\left[Q\right]}^{0}$$

The displacement of the incisal tip from P to Q is achieved through a transformation $T_{R}^{R\prime }$, which includes rotation $\left(\theta_{x},\theta_{y},\theta_{z}\right)$ and translation $\left(t_{x},t_{y},t_{z}\right)$ relative to the right condyle ${\left[x_{R},y_{R},z_{R}\right]}^{T}$. So, we have the following:(6)$${\left[Q\right]}^{R}=T_{R}^{R\prime }{\left[P\right]}^{R}$$
(7)$${T_{R}^{R^{\prime }}=Tran\left(t\right)Rot(\theta_{z},Z^{R})}^{\;}Rot(\theta_{y},Y^{R})Rot(\theta_{x},X^{R})$$
(8)$${Tran\left(t\right)}^{\;}=\left[\begin{array}{cccc} 1 & 0 & 0 & t^{x}\\ 0 & 1 & 0 & \;t^{y}\\ 0 & 0 & 1 & t^{z}\\ 0 & 0 & 0 & 1 \end{array}\right]$$
(9)$${Rot(\theta_{z},Z^{R})}^{\;}=\left[\begin{array}{cccc} cos\theta_{z} & {-sin\theta }_{z\;}\; & 0 & 0\\ sin\theta_{z} & cos\theta_{z} & 0 & 0\\ 0 & 0 & 1 & 0\\ 0 & 0 & 0 & 1 \end{array}\right]$$
(10)$${Rot(\theta_{y},Y^{R})}^{\;}=\left[\begin{array}{cccc} cos\theta_{y} & 0 & {sin\theta }_{y} & 0\\ 0 & 1 & 0 & 0\\ -sin\theta_{y} & 0 & {cos\theta }_{y} & 0\\ 0 & 0 & 0 & 1 \end{array}\right]$$
(11)$$Rot(\theta_{x},X^{R})=\left[\begin{array}{cccc} 1 & 0 & 0 & 0\\ 0 & cos\theta_{x} & {-sin\theta }_{x} & 0\\ 0 & sin\theta_{x} & {cos\theta }_{x} & 0\\ 0 & 0 & 0 & 1 \end{array}\right]$$

Combining Equations (4)–(6), we have the following:(12)$${\left[Q\right]}^{0}={T_{0}^{R}}^{-1}T_{R}^{R^{\prime }}T_{0}^{R}{\left[P\right]}^{0}$$

The lateral excursion will move its position following the condylar path to ${{\left[x_{R}^{\prime },y_{R}^{\prime },z_{R}^{\prime }\right]}^{T}=[x_{R}+t_{R}^{x},y_{R}+t_{R}^{y},z_{R}+t_{R}^{z}]}^{T}$. Furthermore, this lateral excursion will induce a rotation angle $\theta_{y}$ around the $y_{R}$ axis: (13)$$\left\{\begin{array}{c} t_{R}^{x}=f_{R}\left(y_{R}+t_{R}^{y}\right)-x_{R}\\ t_{R}^{z}=g_{R}\left(y_{R}+t_{R}^{y}\right)-z_{R} \end{array}\right.$$
(14)$$\theta_{y\;}={tan}^{-1}\left(\frac{\left|z_{L}^{\prime }-z_{R}^{\prime }\right|}{\left|x_{L}^{\prime }-x_{R}^{\prime }\right|}\right)$$

From Equations (12)–(14), we derive a set of 6 constraint equations to solve the 6-degree-of-freedom problem for ${\left(\theta_{x},\theta_{y},\theta_{z},t_{x},t_{y},t_{z}\right)}_{R}$, using the Newton–Raphson numerical method [29]. The same approach can be employed to obtain ${\left(\theta_{x},\theta_{y},\theta_{z},t_{x},t_{y},t_{z}\right)}_{L}$ for the left lateral excursion by the inherent symmetry of the articulator mechanism.

### 2.2. Virtual Facebow Transfer

In the traditional (mechanical) articulator procedure, ensuring the correct positional relationship between the articulator hinge axis and the dental mold in space requires transferring the maxillary dental mold to the articulator using a facebow. Similarly, in the virtual articulation process, it is also necessary to transfer this relationship correctly to the virtual articulator in order to simulate articulating motion that is consistent with the human body’s motion. The mechanical facebow mechanism consists of a body defining a reference plane, a bite fork, and a bracket structure connecting the bite fork and the facebow body, as shown in Figure 2a. The reference plane passing through the intercondylar axis is determined by two rear reference points and a front reference point. While the shape of the facebow may vary between different brands, the placement of the reference points typically follows a set of rules. The two rear reference points correspond to the centers of the two condylar balls. The front reference point may be either the inferior orbital point or the nasal alar point. The reference plane passing through the inferior orbital point is known as the Frankfurt horizontal plane, while the Camper’s plane passes through the nasal alar point (refer to Figure 2b).

In virtual articulation, the reference plane was defined similarly to the traditional facebow transfer, using two rear reference points and one front reference point based on CT data. In this study, either the Frankfurt plane or Camper’s plane can be used as the reference plane [30]. If the Camper’s plane is chosen, the nasal alar border point serves as the anterior reference point, while the left and right condyles serve as the posterior reference points, as shown in Figure 2c. This method directly utilizes the actual condyles of the human body as the posterior reference points, which makes the line connecting the two posterior reference points the hinge axis of the mandible.

### 2.3. CT Imaging

In order to obtain the CT images that include the complete jaw and TMJ information, we use SIEMENS Emotion 6 (Pittsburgh, PA, USA) (Figure 3a) to capture patient-specific CT data. By adjusting the threshold, the skin and the entire skull were extracted from the CT image. The surface mesh model was then generated using the Marching Cubes algorithm [31] and registered to a color facial 3D scan (RayFace by Ray Inc., Gyeonggi-do, Korea). This contoured face model makes the simulation more realistic. Through the iterative closest point (ICP) registration [32] of the upper and lower jaw models and the scanned teeth model, the segmented teeth of the CT data were replaced with high-precision scanned teeth. During the CT imaging, the patient was specially required to clench the upper and lower jaws to the maximum intercuspation (MIP). This ensured that the digital dental model was positioned according to this occlusal relationship as the initial position of the jaw movement tracking. The integrated result is shown in Figure 3b.

### 2.4. Tracking of Patient-Specific Jaw Movement

In this research, a dual-marker optical tracking system including an eyeglass frame combined with the tracking marker and a 3D-printed splint was used to track the relative movement of the upper jaw and lower jaw, respectively (Figure 4). Before tracking, let us first define the articulator (world) coordinate system of the virtual articulator as $\left\{X^{0},\;Y^{0},\;Z^{0}\right\}$, the coordinate system of the lower marker as $\left\{X^{low},\;Y^{low},\;Z^{low}\right\}$, the coordinate system of the upper marker as $\left\{X^{up},\;Y^{up},\;Z^{up}\right\}$, and the camera coordinate system as $\left\{X^{cam},\;Y^{cam},\;Z^{cam}\right\}$. Since the lower marker is attached to the lower jaw, its movement can be regarded as a rigid body movement as opposed to the upper jaw. If a point on the lower jaw is denoted as ${\left[P\right]}^{low}$, its relationship to the coordinate system of the virtual articulator can be expressed as Equation (15). Here, $T_{cam}^{low}$ and $T_{cam}^{up}$ represent the coordinate transformations between the camera and the lower and upper markers, respectively. The optical measurement and the calculation of the transformation matrices can be obtained by the method proposed in our previous publication [24].
(15)$${\left[P\right]}^{low}={T_{cam}^{low}\cdot T}_{up}^{cam}{\cdot T}_{0}^{up}\cdot {\left[P\right]}^{0}$$

To register the tracking markers to the virtual articulator coordinate system, we used 3D scanning to scan the human face with the upper tracking marker. The human face was matched with the CT skin surface mesh using the ICP algorithm [32], thereby obtaining the relationship $T_{0}^{up}$ between the upper marker coordinate system and the virtual articulator coordinate system, as shown in Figure 5. The position of the lower marker in the virtual articulation coordinate system can be obtained indirectly from the relationship between the two markers and the camera. When the lower jaw moves, a point P on the lower jaw moves to $P^{\prime }$. Since the lower jaw and the lower marker are assumed to be a rigid body, $P^{\prime }$ can be deduced from Equation (16) in the virtual articulation coordinate system.
(16)$${\left[P^{\prime }\right]}^{0}=T_{up}^{0}\cdot T_{cam}^{up}\cdot T_{low}^{cam}{\cdot \left[P^{\prime }\right]}^{low}$$

Through the above derivation, we can track the movement of any point on the lower jaw, in which the movement trajectory of the condyles and the incisal edge of the incisor can be used as the information required for personalized articulation. Since the tracking system uses the calculation of the relative movement of the upper and lower markers, it can reproduce the relative movement between the upper and lower jaws.

### 2.5. Verification Method

The accuracy of simulation and modeling can be verified by comparing simulation data to measurement data. In this work, we aim to evaluate the accuracy of the jaw motion mathematical model by estimating differences between the simulated trajectory of jaw motion and the (measured) real trajectory. The real trajectory is obtained by optically measuring the pose of the lower jaw (including rotation and translation) to obtain the condyle positions using lower jaw geometry from CT/CBCT. Meanwhile, the simulated trajectory is obtained by using the mathematical articulation model to inversely calculate the condyle positions from the measured incisor edge points. Various error indices can be used to estimate the difference between the two trajectories using sampled discrete data points. One commonly used error index is the average trajectory error (ATE), which can be expressed by the following equation [33]:(17)$${ATE}_{all}=\sqrt{\frac{1}{N}\sum_{i=1}^{N}{\left\|p_{i}-f^{-1}(q_{i})\right\|}^{2}}$$
where $p_{i}$ is the measured condyle position, $q_{i}$ is the measured incisor edge point, and $f^{-1}(q_{i})$ is the simulated condylar position through the inversed jaw motion model.

To verify the accuracy of the personalized mathematical model of jaw movement derived in this study, a set of experimental methods was designed for accuracy verification analysis. The principle of this verification method is to compare the trajectory of the left and right condyles with the trajectory of the simulation when the jaw moves. Initially, posture estimation of the tracking marker during the mandible movement was used to derive the related positions of the condyles, thereby obtaining a moving trajectory of the left and right condyles. Then, the recorded position of the incisal edge was used in conjunction with the mathematical model of mandibular movement derived in this study to inversely calculate the corresponding left and right condyle positions. Consequently, a simulated movement trajectory of the left and right condyles was obtained. Finally, the accuracy indicator was obtained by comparing the similarity of the two curves. The process is illustrated in Figure 6.

In computer vision and computational geometry fields, the Fréchet distance [34] is commonly used to describe the similarity or discrepancy between paths. As the condylar curve derived from measurements constitutes a polygonal chain with discrete points, this study employs the discrete Fréchet distance [35] to evaluate the discrepancy between the measured trajectory and the simulated trajectory obtained from the proposed virtual articulation model. Thereby, we can estimate the accuracy of the virtual articulation model developed in this study.

## 3. Results

### 3.1. Registration of Jaws to the Articulation Coordinate System

The digital teeth model was positioned within the virtual articulator coordinate system in accordance with human anatomy, achieved through the registration of the digital teeth model with the cranial CT model. The condyles of the virtual articulator were matched with the corresponding human condyles identified in the CT data. Given the nearly spherical shape of the condyle, a spherical fitting approach was employed, utilizing a ball to conform to the condyle’s morphology. The centroids of the fitted balls were then employed as posterior reference points.

In this work, a user interface was implemented to enable users to select reference points on the cranial CT model, as illustrated in Figure 7. Subsequently, employing the computation of a transformation matrix, the spatial alignment of the cranial CT model and the digital teeth model within the virtual articulator coordinate system was achieved. 

### 3.2. Motion Tracking of Upper and Lower Jaws

Following the affixation of two fiducial markers onto the upper and lower jaws, the patient was required to adjust the occlusal relationship to the centric occlusion (CO) position as the initial position of the tracking. Once confirming that the two markers are within the working range of the camera, the tracking can be started. The tracking results were subsequently combined with the cranial surface mesh model. Figure 8 visually depicts the real-time observation of jaw movement and the concurrent trajectory of the condyle. Additionally, the camera-captured image is displayed in the upper right corner of the screen.

In particular, the movement trajectory of the condyles during the protrusion and lateral excursions can be recorded. The tracking outcomes allow physicians to diagnose TMJ disorders during mandibular movement. Furthermore, these results can be used to personalize the condylar path of the virtual articulator.

### 3.3. Simulation of Virtual Articulation

In this work, a computer simulation program was built based on the personalized mathematical model of jaw movement, which can perform dynamic occlusion simulation and display the patient’s upper and lower jaws, condylar paths, and interference points during occlusion. This simulation program provides useful information to assist dentists in designing dentures or other dental-related products. Its interface is shown in Figure 9a.

The system moves the incisal point according to the incisal guidance path and calculates the posture of the mandible at different positions on the path. However, the motion path is programmable based on our mathematical model and changes with the collision of the upper and lower jaw teeth. For interference checking, we employ sphere collision detection and adjust the motion path according to the interference between teeth during the motion. Therefore, this approach enables users to design new prostheses effectively. By exploiting the virtual articulator, they can simulate the patient’s occlusal path during forward, left, and right protrusions, as shown in Figure 9b–f. Subsequently, the simulation results provide valuable insights for crafting a proper prosthetic design with the personalized occlusal dynamic.

### 3.4. Personalized Condylar Path

Most mechanical articulators adopt a fixed straight-line or a curve pattern for representing the condylar path. However, the condylar path is in fact a personalized 3-dimensional trajectory. In this work, to combine the mandibular motion information with the mathematical model of the articulation, the condylar path obtained by the above tracking method was fitted and combined with the mathematical model of the articulation. The articulation has two adjustment parameters that represent the shape of the condyle projection in the horizontal and sagittal planes [36]. Therefore, by projecting the tracked left and right condyle paths to the horizontal and sagittal planes, we fitted a condylar path corresponding to the articulation. The methodology is detailed as follows:


The points on the condylar path of the protrusion process were projected onto the sagittal plane and fitted with a quadratic curve, which contains the personalized condyle shape and condylar inclination;At the same time, the condylar path on the working side was projected onto the horizontal plane and fitted with a straight line. This linear trajectory represents the movement of the condyle during lateral motion on the working side, corresponding to the Bennett angle of the articulation, as shown in Figure 10;The curve equations of these two two-dimensional planes can be combined into a three-dimensional curve in space, as shown in Equation (18).

(18)
$$C\left(u\right)=\frac{\sum_{j=0}^{n}N_{j,p}(u)w_{j}\overset{⃑}{P_{j}}}{\sum_{j=0}^{n}N_{j,p}(u)w_{j}}$$



This 3-dimensional curve (personalized condylar path) can be determined by a curve fitting regression process. The result can be represented as a parametric B-Spline or NURBS (Non-Uniform Rational B-Spline) curve in 3D space, as shown in Figure 11a.

This personalized condylar curve equation replaces the fixed articulator mechanism of traditional mechanical articulators, so that a personalized mathematical model of jaw movement can be obtained. When tracking the mandible, the incisal edge movement trajectory of the patient’s protrusion and lateral excursions are also recorded, as shown in Figure 11b. These occlusal paths are used as the movement direction of the articulation, aligning the incisal movement to guide the occlusal path according to the patient’s habitual movement. 

### 3.5. Verification and Analysis of the Jaw Movement

Experiments were conducted on the occlusal paths of anterior protrusion and lateral excursions. Each movement was conducted from the CO position to the extreme position, back and forth twice. After adding the mandibular motion information to the mathematical model of the articulator, we imported the positions of the incisal edge to calculate the positions of the condyles. The data were analyzed to compare the condylar trajectories between the direct tracking measurement and the inverse calculation from the incisal edge. The trajectory comparison results are presented in Figure 12, where the blue line represents the simulated trajectory, and the red line represents the tracking trajectory.

The discrete Fréchet distances [34] for each occlusal movement are shown in Table 1.

Based on the results of the experiment, we found that the Fréchet distance between the tracked mandible movement and the simulated condyle movement in this study was approximately 1.7 mm.

## 4. Discussion

Our study provides a comprehensive digital workflow for developing a personalized virtual articulator using CT scans and jaw movement data from patients. The results verify that the constructed mathematical model effectively controls mandibular movement without the need for a mechanical or digital articulator. From raw CT scan data, we successfully segmented the upper and lower jaws for subsequent registration. Through the registration of the digital teeth model with the cranial CT model, the digital teeth model was positioned within the virtual articulator coordinate system. As a result, the condyles of the virtual articulator were matched with the corresponding human condyles. The jaw motion tracking allowed for recording the movement trajectory of the condyles during protrusion and lateral excursions. This capability opens up applications for the developed virtual articulator in TMJ disorder diagnosis. The developed software program based on available libraries such as C++ and OpenGL enables simulating the positions of occlusal paths on patients. The personalized condylar curve obtained serves as a substitute for the rigid articulator mechanism inherent in conventional mechanical articulators. This innovation allows the derivation of a customized mathematical representation of jaw movements. Furthermore, the tracked mandible movement and the simulated condyle movement were verified through an acceptable Fréchet distance of 1.7 mm to demonstrate the feasibility of the proposed model.

Nowadays, there are commercial digital articulators such as the Ceramill Map 300 CAD/CAM system by Amann Girrbach (Koblach, Austria) [6], Dental CAD/CAM System by 3Shape (New Providence, NJ, USA) [7], and exocad articulator by Exocad GmbH (Darmstadt, Germany) [37]. However, most commercial systems today only simulate the mechanical articulator behavior in general, and do not take into account the patient’s individual CT information for customized articulation. Furthermore, commercial systems often do not have verification methods to show the accuracy of the simulation. Therefore, it is questionable whether they can be truly reliable or accurate in determining the correct occlusion when it comes to designing dental prostheses.

Every individual possesses a unique articulation and occlusion. To personalize individual articulation, a mathematical articulation model and the following personalized parameters must be determined: Bennett angle [36], condyle inclination angle [38], and condylar path [39]. C++ and OpenGL libraries were applied to develop the virtual articulator, and used VTK and ITK libraries to develop jaw extraction software [40]. In order to obtain comprehensive information on the patient’s jaw movement, it is necessary to take a CT image of the patient’s head and scan the teeth. Each data set needs to undergo a series of registrations due to its different coordinate systems.

There are several commercial jaw-motion tracking systems that use high-resolution optical scanning devices, such as Zebris (Isny, Germany) [41] and Modjaw (Villeurbanne, France) [42]. While these state-of-the-art commercial systems excel in recording the relative jaw motion between the upper and lower jaw, they rely on fixed kinematic mechanisms of mechanical articulators rather than patient-specific condylar paths for occlusal analysis. Despite the tedious registration process required by systems like Zebris, which utilize analog transfer devices like metal para occlusal spoons for the lower jaw transfer and plastic coupling spoons for the upper jaw transfer, commercial systems like Modjaw have not integrated CT/CBCT information to connect jaw motion to digital dental models. These sequential and separated solutions from jaw tracking to dental CAD and then to DICOM viewer result in cumbersome registration and connection processes that impede workflow efficiency. In short, current commercial systems are not fully integrated, but rather, they are operated individually with intermediate steps that require additional time and effort from dentists and patients alike.

In contrast, our proposal integrates a mathematical jaw motion model that combines CT data analysis to identify and register anatomical/geometric attributes, such as condylar distance, reference planes, and patient-specific condylar paths. This integrated information provides an individualized jaw motion model that can be used for programmable jaw motion and occlusal analysis. Simulation results from the software built based on the mathematical model demonstrate its capability to work accurately with various mandible postures. The main difference between our proposal and systems like Zebris or Modjaw is that their occlusion analysis is based on either recorded jaw motion paths or the kinematic analysis of lab-based digital mechanical articulators. In other words, their occlusal analysis still relies on lab-based mechanical articulators but is digitally implemented. The choice of the type of mechanical articulator used will influence the occlusion analysis results significantly. Conversely, our proposal is based on a generalized mathematical jaw motion model that accounts for both static patient anatomy (from CBCT) and dynamic patient-specific condyle movements. Our model can program the jaw motion dynamically to satisfy various functional requirements such as occlusion analysis or splint design. Additionally, compared to the hinge axis arbitrarily established by the traditional facebow, this study’s definition aligns more closely with the human anatomy structure. With the CT data-defined reference plane, aligning the human skull and virtual articulator becomes easier without relying on the traditional mechanical facebow. 

Artificial intelligence (AI) applications in the medical field in general, and dentistry in particular, have become increasingly prevalent and providing substantial value over the past two decades [42,43]. With the computational power combined with intelligent algorithms, AI has permeated various aspects, ranging from diagnosis and treatment to simulation [44,45]. In this study, we employed AI to segment the upper and lower jaw for registration based on the ICP algorithm. However, AI-related applications can be implemented to simplify operations and enhance the precision of the current work. Specifically, the matching process of condyles between the developed virtual articulator and corresponding human (CT data) was performed manually. By selecting a few points on the condyle region, an imaginary sphere was formed to imitate the morphology and represent the condyle. This process is practically feasible for AI applications with automaticity and high accuracy. A study has demonstrated that TMJ segmentation can be achieved on CBCT images with an intersection over union (IoU) of up to 0.955 for condyles [46]. Furthermore, AI segmentation assists in restructuring the 3D model of teeth [47], making improvements in the design and production of dental equipment. Alongside the availability of current manufacturing methods, notably additive manufacturing, future fabrication of products such as splints and aligners is conceivable [48,49]. 

The Fréchet distance result indicates an average deviation of the condyle position during the simulation of occlusal movement at 1.7 mm, accounting for errors from intra-oral scans, CT scans, registration, marker tracking, and mathematical modeling. The limitations of the laboratory equipment with low accuracy contribute to the overall error in this study, which is reflected in the value of the Fréchet distance. While this research demonstrates the feasibility and effectiveness of the proposed solution in the university research environment and laboratory setting, there is room for improvement in accuracy. This could be achieved by employing higher-resolution cameras or commercial optical tracking devices. As modern camera technology continues to advance, we anticipate an enhancement in the accuracy of jaw tracking. In this study, we focused on building a digital workflow for the development of a personalized virtual articulator, without delving into extensive clinical applications. Nevertheless, the methodology involved experiments on human samples, utilizing CT scan images and capturing real jaw movement. Along with reported results, the developed virtual articulator demonstrates to be feasible for various clinical applications such as TMJ disorder diagnosis [50] and occlusion analysis [23]. The ability to track the movement trajectory of the condyles and gather information on occlusal paths from simulations allows for the straightforward and accurate detection of TMJ disorder signs and facilitates the creation of a proper prosthetic design.

## 5. Conclusions

This research work presents the development of a personalized virtual articulator constructed from CT data and jaw movement information. Such an approach enables the creation of a unique mathematical model of dental articulation, which can be utilized by dental professionals in prosthesis design and treatment planning. With this method, digital simulation of patient-specific occlusion can be performed remotely, even when the patient is not present in the clinic.

## Figures and Tables

**Figure 1 bioengineering-10-01248-f001:**
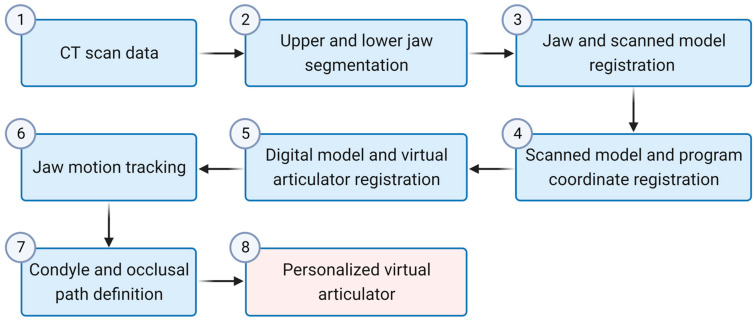
Flow chart of constructing personalized articulation.

**Figure 2 bioengineering-10-01248-f002:**
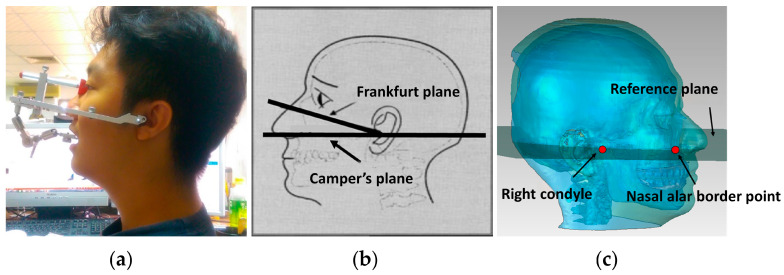
Setup of virtual facebow transfer: (**a**) Mechanical facebow; (**b**) Frankfurt and Camper’s planes; (**c**) the Camper’s plane using CT data.

**Figure 3 bioengineering-10-01248-f003:**
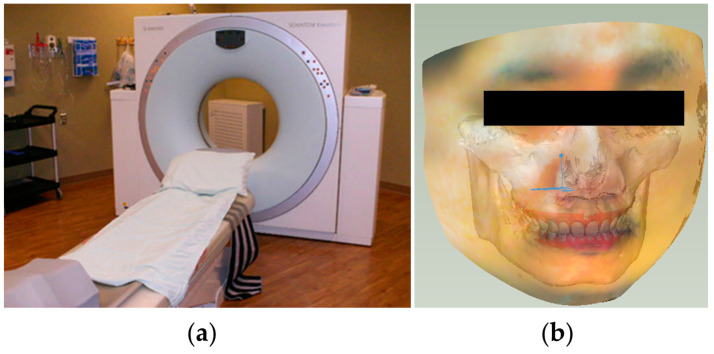
CT imaging: (**a**) SIEMENS Emotion 6; (**b**) Digital avatar of a virtual patient.

**Figure 4 bioengineering-10-01248-f004:**
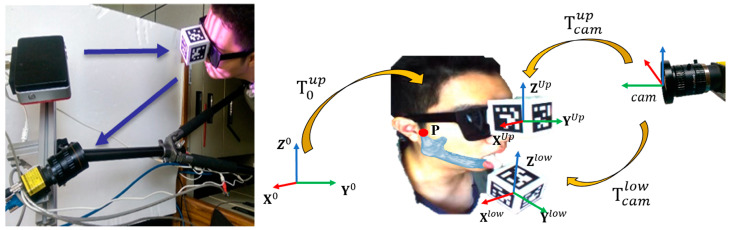
Setup of the tracking system.

**Figure 5 bioengineering-10-01248-f005:**
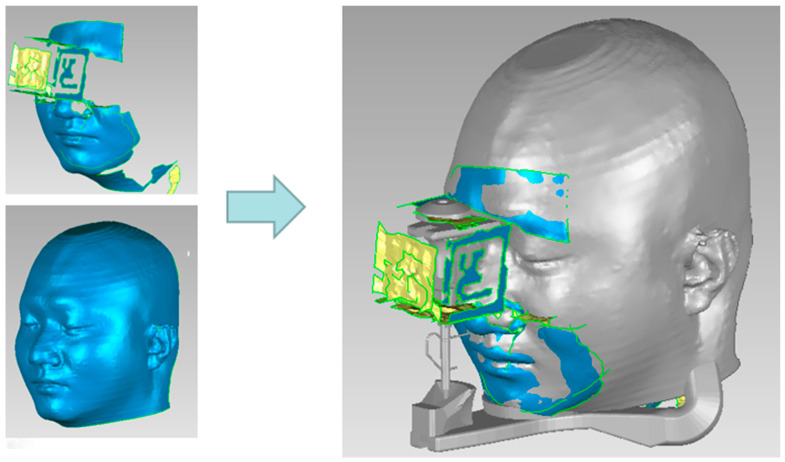
Registration between the upper marker and the virtual articulation coordinate system.

**Figure 6 bioengineering-10-01248-f006:**
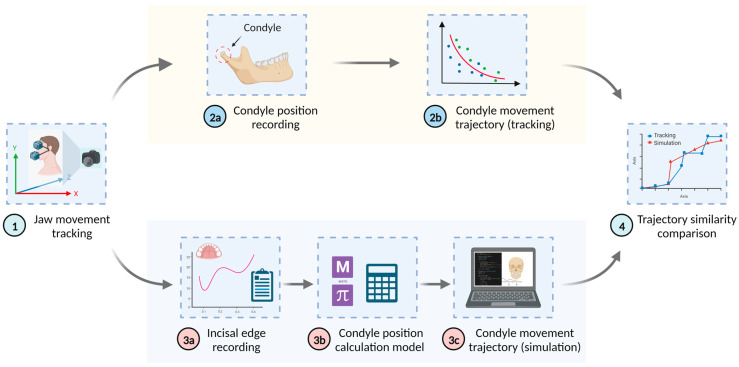
Flow chart of jaw movement verification and analysis. Step 1 involves tracking data to obtain information on the condyle position and incisal edge for tracking (step 2) and simulating (step 3), respectively. The obtained results of the condyle movement trajectory were compared in step 4.

**Figure 7 bioengineering-10-01248-f007:**
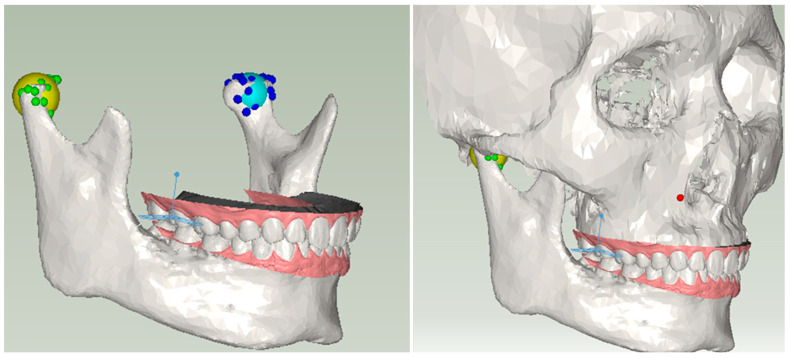
Setup of the three reference points. The right and left condyles in the virtual articulator are represented by two corresponding spherical models in yellow and blue using a spherical fitting approach.

**Figure 8 bioengineering-10-01248-f008:**
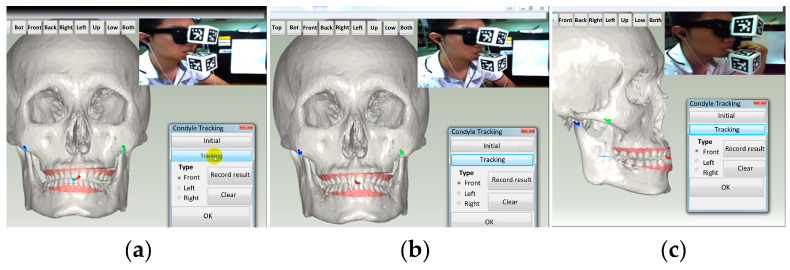
Real-time tracking. (**a**) Right excursion; (**b**) Left excursion; (**c**) Protrusion.

**Figure 9 bioengineering-10-01248-f009:**
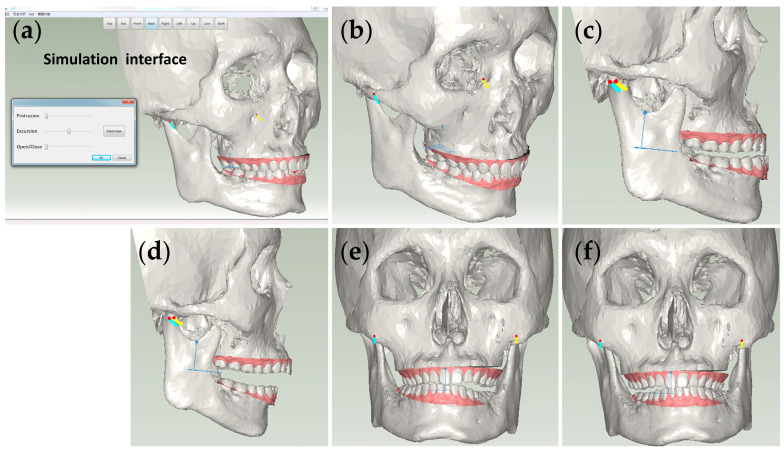
Simulation of virtual articulation: (**a**) User interface of virtual articulation; (**b**) CO position; (**c**) Protrusion; (**d**) Opening; (**e**) Right excursion; (**f**) Left excursion.

**Figure 10 bioengineering-10-01248-f010:**
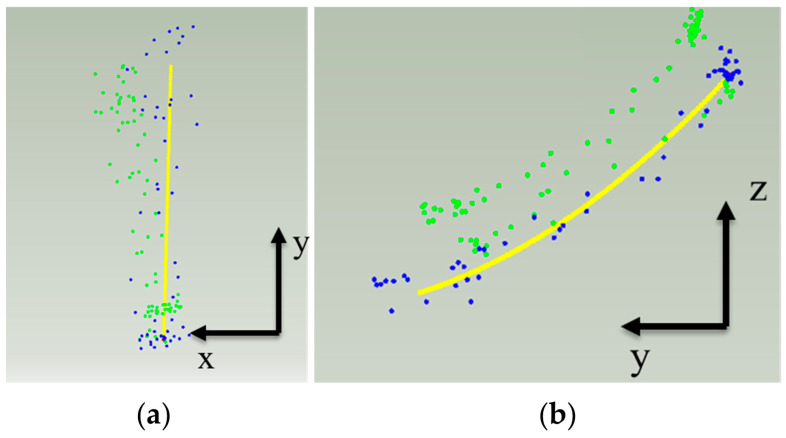
Projection of condylar path on (**a**) horizontal plane and (**b**) sagittal plane, in which blue points represent protrusion and green points represent lateral excursion.

**Figure 11 bioengineering-10-01248-f011:**
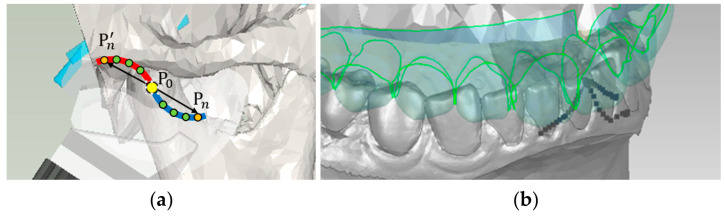
Personalized condylar path: (**a**) Condylar path represented by a parametric curve; (**b**) Recorded occlusal paths by tracking of the incisal edge.

**Figure 12 bioengineering-10-01248-f012:**
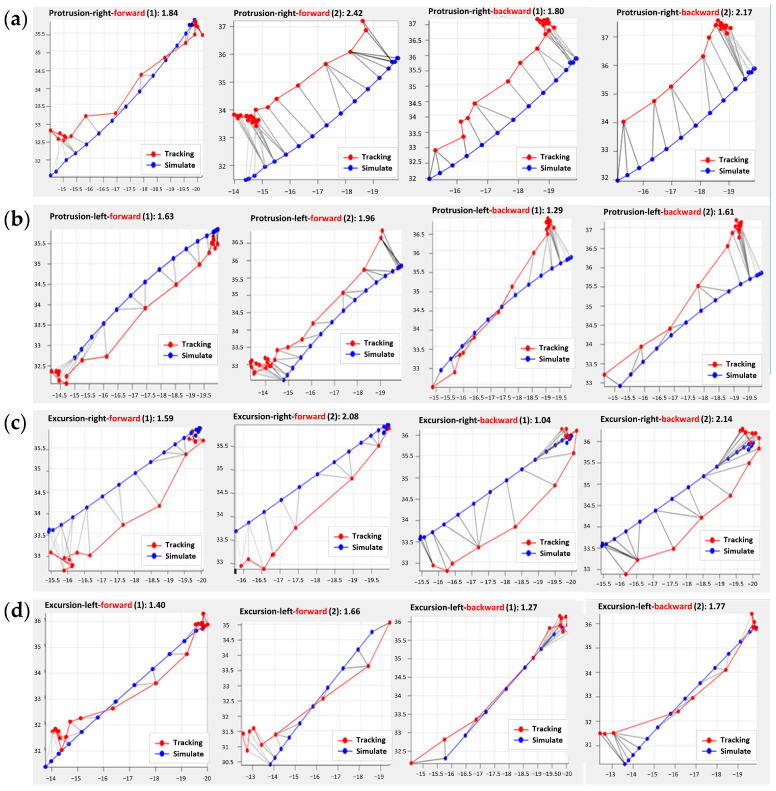
Verification of protrusion condylar path: (**a**) Right condyle and (**b**) left condyle. Verification of lateral excursion path: (**c**) Right condyle and (**d**) left condyle.

**Table 1 bioengineering-10-01248-t001:** Similarity comparison of the jaw movement: tracking versus simulation.

Protrusion	Right Condyle	Left Condyle
Forward	1.84	1.63
	2.42	1.96
Average	2.13	1.79
Backward	1.80	1.29
	2.17	1.61
Average	1.98	1.45
**Right excursion**	**Right condyle**	**Left condyle**
Forward	1.59	1.40
	2.08	1.66
Average	1.83	1.53
Backward	1.04	1.27
	2.14	1.77
Average	1.59	1.52

Unit: mm.

## Data Availability

Data is available upon reasonable request to the correspondence author.

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
