# Peer review of "Virtual Dental Articulation Using Computed Tomography Data and Motion Tracking"

_bioengineering, 2023, doi:10.3390/bioengineering10111248_

Round 1

Reviewer 1 Report

Dear Authors,

I’ve extensively read the manuscript titled “Virtual Dental Articulation Using Computed Tomography Data 2 and Motion Tracking.”. The aim of this study is to presents the development of a virtual articulator capable of reproducing personalized jaw movements and a digital workflow of creating virtual articulation, utilizing jaw movement and information obtained from CT scans. The methodology is appropriate and quite linear with recent evidences/ studies on this topic. I’ve not major concerns in this regard.

Some aspects must be addressed before considering the manuscript suitable for publication:

-       A significant revision of scientific English language is required.

-       Authors should briefly argue about the possible implication of 3D printing technology of the proposed method, citing appropriate reference in terms of accuracy https://pubmed.ncbi.nlm.nih.gov/35792192/

https://pubmed.ncbi.nlm.nih.gov/34100158/

Authors should briefly argue about the possible implication of artificial intelligence applied on anatomical segmentation on the proposed method, citing appropriate reference https://pubmed.ncbi.nlm.nih.gov/34553817/

-       In general, the discussion section should be improved, even considering the above mentioned topics

-       Was power analysis performed?

significant english language revision is required

Author Response

Thank you very much for taking the time to review this manuscript. We highly appreciate the feedback, and based on it, necessary revisions have been made. Please find the detailed responses and the corresponding revisions/corrections highlighted/in track changes in the response and re-submitted files. 

Reviewer 2 Report

This article proposes a practical solution to the typical approaches to the modeling of complex jaw movements in order to obtain diagnostic information about the TMJ and configure individual settings of occlusal analysis for patients in the dental field. The description of the theoretical approach and the technical solutions is very accurate and convincing. For these reasons I strongly suggest to publish the article in its present form.

Author Response

Thank you very much for taking the time to review this manuscript. We are delighted and appreciate your positive evaluation of our work!

Reviewer 3 Report

Dear authors, I find the manuscript very interesting, but some modifications are necessary. Lines 82-96 are not well positioned in the Introduction section. The aim of the study must be highlighted.  It is not usual to finish the Introduction with the sentence: IN conclusion,,, (line 92). There is a lot of text in the Results that should be moved to the Material and Methods.  The conclusions are not supported by the results The limitations of the study should be added as well as the clinical implications. My suggestion is to rearrange the manuscript and rewrite it.

Author Response

(The authors gave the same response as above.)

Round 2

Reviewer 3 Report

Dear authors, thank you very much for accepting my suggestions.